# Incomplete Decarboxylation of Acidic Cannabinoids in GC-MS Leads to Underestimation of the Total Cannabinoid Content in Cannabis Oils Without Derivatization

**DOI:** 10.3390/pharmaceutics17030334

**Published:** 2025-03-05

**Authors:** Martina Franzin, Rebecca Di Lenardo, Rachele Ruoso, Riccardo Addobbati

**Affiliations:** Institute for Maternal and Child Health, IRCCS “Burlo Garofolo”, Via dell’Istria 65/1, 34137 Trieste, Italy; rebeccadilenardo@gmail.com (R.D.L.); rachele.ruoso@burlo.trieste.it (R.R.); riccardo.addobbati@burlo.trieste.it (R.A.)

**Keywords:** cannabis oil, cannabinoids, decarboxylation, acid forms, GC-MS, LC-MS

## Abstract

**Background**: Cannabis oil titration consists of quantification of the acidic precursors tetrahydrocannabinolic acid (THCA) and cannabidiolic acid (CBDA) and their decarboxylated products, the active neutral cannabinoids delta-9-tetrahydrocannabinol (Δ9-THC) and cannabidiol (CBD), and is recommended to ensure galenic preparation quality through gas and liquid chromatography coupled with mass spectrometry (GC-MS; LC-MS). Analyses by LC-MS and GC-MS involving derivatization allow for detection of acidic and neutral cannabinoids; on the contrary, GC-MS without derivatization determines only neutral cannabinoids due to high temperature-decarboxylation occurring in the injection system. However, it is not clear if decarboxylation is complete. **Methods**: Different GC-MS methods with (BSTFA: TMCS and pyridine; incubation at 60 °C for 25 min) or without derivatization and an LC-MS method were developed for cannabinoid quantification. The total Δ9-THC and CBD yield of recovery were compared between the methods by testing laboratory samples with known concentrations of THCA and CBDA (total Δ9-THC and CBD: 175–351–702 ng/mL) and real cannabis oil samples (n = 6). **Results**: The total Δ9-THC and CBD yield of recovery were determined using LC-MS and GC-MS with derivatization, but not using GC-MS without derivatization (decarboxylation conversion rate of about 50–60%). No high deviation (<10%) in the total neutral cannabinoid concentrations in real cannabis oil samples was noticed, probably due to the low content of acidic forms in the original galenic preparation. **Conclusions**: This study raised awareness about the potential underestimation of the total Δ9-THC and CBD content in cannabis oils when quantification is performed by GC-MS without derivatization. The advice for pharmacists is to perform complete decarboxylation to convert all acidic precursors in neutral cannabinoids.

## 1. Introduction

The use of *Cannabis sativa* preparations is known to be therapeutically relevant in several clinical disorders affecting both adult and pediatric populations [1,2]. In particular, as regulated by the Ministerial Decree of 9/11/2015 in Italy, cannabis preparations are administered under medical prescription as part of therapeutic regimens for chronic pain related to multiple sclerosis; nausea and vomiting induced by chemotherapy, radiotherapy, and antiretroviral therapy; appetite stimulation in cancer and AIDS patients with anorexia; reduction of intraocular pressure in glaucoma; and involuntary movements reduction in Gilles de la Tourette syndrome [1,3,4,5,6]. Notably, even if not yet approved in Italy, several trials have also highlighted the application of medical cannabis in the pediatric field, particularly for drug-resistant epilepsy, including Dravet syndrome, demonstrating suitable efficacy and safety profiles [2,7,8,9].

Among the 500 identified constituents of the *C. sativa* phytocomplex (including sesquiterpenes, sugars, hydrocarbons, steroids, flavonoids, nitrogenous compounds, and amino acids), C_21_ terpeno-phenolic cannabinoids are considered the most relevant from a pharmacological point of view [3,10,11]. In particular, the main active compounds are delta-9-tetrahydrocannabinol (Δ9-THC) and cannabidiol (CBD); the former is responsible for the psychotropic effects of *C. sativa* preparations, while the latter has anti-inflammatory and antioxidant properties [12,13]. Depending on the therapeutic need, different preparations derived from *C. sativa* varieties, which vary in the Δ9-THC and CBD content, are prescribed [3]. *C. sativa* varieties produced in Italy, such as FM1 (Δ9-THC: 13–20% *w*/*w*; CBD < 1% *w*/*w*) and FM2 (Δ9-THC: 5–8% *w*/*w*; CBD: 7.5–12% *w*/*w*), have become available since the Ministerial Decree of 9/11/2015 authorized the cultivation of medical cannabis by the Military Chemical and Pharmaceutical Works of Florence (ICFM) [3,4,14]. Furthermore, given the high consumption (estimated at 860 tons of raw material in 2019), *C. sativa* varieties, such as Bedrocan (Δ9-THC: 22% *w*/*w*; CBD < 1% *w*/*w*), Bedrobinol (Δ9-THC: 13.5% *w*/*w*; CBD < 1% *w*/*w*), Bediol (Δ9-THC: 6.5% w/w; CBD: 8% *w*/*w*), and Bedrolite (Δ9-THC: < 1% *w*/*w*; CBD: 9% *w*/*w*), are imported from the Netherlands [3,15].

Beyond cannabis-based herbal medicines authorized on the Italian market, galenic preparations, such as decoctions and oil extracts, are also approved [3,16,17,18,19,20]. In particular, cannabis oil represents an interesting pharmaceutical form given the bioavailability of cannabinoids after extraction in this lyophile matrix; therefore, Italian law (9/11/2015, 279 Official Gazette) recommends the titration of active compounds through sensitive and specific methodologies to ensure the quality of the galenic preparation [3,4,20].

Cannabinoids are biosynthesized primarily as carboxylic acids and stored in the glandular trichomes of flower bracts [21]. Particularly, the acidic forms tetrahydrocannabinolic acid (THCA) and cannabidiolic acid (CBDA) are initially the most abundant and can be converted into the neutral and active cannabinoids Δ9-THC and CBD through a non-enzymatic decarboxylative process, which takes place in specific conditions of temperature, light, and oxygen [22]. Hospital and local pharmacists prepare cannabis oil using different extraction methods, which mainly involve the decarboxylation of acidic forms into neutral ones [20,23]. Subsequently, quantitative analyses using sensitive and specific methods such as gas chromatography or liquid chromatography coupled with mass spectrometry (GC-MS, LC-MS) are performed [24,25]. Both neutral and acidic cannabinoids are identified by performing LC-MS analysis or GC-MS analysis with a derivatization step [26]. In particular, derivatization involving silylation with reagents, such as N,O-bis(trimethylsilyl)trifluoroacetamide (BSTFA), implies the addition of a trimethylsilyl group to the carboxylic group, making acidic cannabinoids more volatile and avoiding decarboxylation that can occur because of the high temperature of the GC injection system [27]. Additionally, GC-MS without derivatization is used to detect only neutral cannabinoids, derived from the ones present in the galenic preparation and from the decarboxylation of acidic forms during injection, resulting in the total Δ9-THC and CBD content in the galenic preparation [26]. To date, it has not been fully elucidated if the acidic forms THCA and CBDA could be completely converted into the neutral and active Δ9-THC and CBD [10,26,28,29].

Interestingly, depending mainly on the extraction and quantification methods, there is wide inter-lot and interlaboratory variability in the total Δ9-THC and CBD content measured in cannabis oils [16,17,20].

In this context, this study aims to evaluate the percentage of conversion of the acidic forms of cannabinoids THCA and CBDA into their corresponding neutral ones using a GC-MS method, which does not involve derivatization. The results will be compared with the ones obtained from a GC-MS method with sample derivatization and an LC-MS method for the quantification of both forms of cannabinoids. The impact on the quantification of real samples of cannabis oil was also evaluated.

## 2. Materials and Methods

### 2.1. Galenic Preparation

Cannabis oil samples (n = 6) used in this study were left over from a diagnostic routine analysis performed at the Advanced Translational Diagnostics Laboratory at IRCCS “Burlo Garofolo” in Trieste. In particular, galenic preparations of cannabis oil were obtained from local pharmacies according to the harmonized extraction methods defined by the Italian Society of Compounding Pharmacists (SIFAP) [23,24]. In particular, the extraction protocol consisted of heating *C. sativa* varieties (high Δ9-THC content: Bedrocan (1); intermediate Δ9-THC content: Bediol (5)) in olive oil (ratio 1:10; 5 g of inflorescences in 50 mL of oil) to obtain decarboxylation and maceration [23,24].

### 2.2. Chemicals and Reagents

All the chemicals and reagents used in this study were of analytical grade. Methanol (≥99.9%), formic acid (≥95%), N,O-bis(trimethylsilyl)trifluoroacetamide with trimethylchlorosilane (BSTFA:TMCS, 99:1, *v*/*v*), and pyridine (purity ≥ 99.6%) were purchased from Merck (Darmstadt, Germany). Ultrapure water was obtained from Biosolve Chimie (Dieuze, France). Analytical standards in methanol, consisting of Δ9-THC (concentration: 1000 μg/mL; purity ≥ 99.2%), CBD (concentration: 100 μg/mL; purity ≥ 99.6%), THCA (concentration: 1000 μg/mL; purity ≥ 99.2%), and CBDA (concentration: 100 μg/mL; purity ≥ 98.5%), were purchased from LGC Standards Srl (Milan, Italy). A certified internal standard mix (IS), consisting of the deuterated compounds of Δ9-THC and CBD (Δ9-THC-d3 and CBD-d3), was provided by Eureka Lab Division (Ancona, Italy).

### 2.3. Working Solutions

All the working solutions reported below were obtained by dilution of cannabinoids standards in methanol. Calibrators (CAL) at known concentrations of both neutral and acidic cannabinoids (n = 6; concentrations of CAL: 0–50–100–250–500–1000 ng/mL) were prepared and used for the LC-MS analysis and the GC-MS analysis including the derivatization step. Furthermore, CALs at known concentrations of only neutral cannabinoids (n = 6; concentrations of CAL: 0–50–100–250–500–1000 ng/mL) were also prepared and used for the GC-MS analysis without the derivatization step. Three levels of quality controls (QC), containing only THCA and CBDA (QCI–QCII–QCIII: 200–400–800 ng/mL), were prepared and tested.

### 2.4. Instrumentation

Analyses of cannabinoids were performed using two different instruments. Particularly, GC-MS analyses required the use of an Agilent 7697A (Agilent Technologies, Milan, Italy) combined with an Agilent 8890 GC/Agilent 5977C GC/MSD (Agilent Technologies, Milan, Italy), while LC-MS analyses were performed using an HPLC Exion LC 2.0 (Sciex, Milan, Italy) combined with a QTRAP 6500+ system (Sciex, Milan, Italy).

### 2.5. Quantification of Neutral Forms of Cannabinoids Using GC-MS Analysis Without Derivatization

#### 2.5.1. Sample Preparation

Fifty µL of the IS mix were added to 100 µL of solutions of CALs and QCs in methanol/cannabis oil previously diluted in methanol 20,000 times. After vortexing, the sample was dried under a gentle nitrogen stream and subsequently reconstituted in 130 µL of ethyl acetate.

#### 2.5.2. GC Conditions

Two μL were injected in the instrument with a split ratio of 5:1. The temperature of the injection port and the transfer line was set at 280 °C. A DB-5MS UI capillary column (30 m × 0.25 mm I.D., 0.25 μm film thickness, with 5% phenylmethylsiloxane) (Agilent Technologies, Milan, Italy) was used to achieve separation. Helium, the carrier gas, was set at a constant flow of 1.2 mL/min for the total run time (30 min). Oven temperature was initially set at 50 °C and rose with a temperature ramp of 10 °C/min until 320 °C; this temperature was held for 3 min.

#### 2.5.3. MS Conditions

The temperatures of the ion source and the quadrupole were set at 230 °C and 150 °C, respectively. The electron ionization mode was positive. The solvent delay was set at 21 min. The mass spectrometer operated in selected ion monitoring (SIM). As such, the ions monitored were 299 and 231 for Δ9-THC; 231 and 246 for CBD; 302 for Δ9-THC-d3; and 234 for CBD-d3. The dwell time was set at 70 ms.

### 2.6. Quantification of Neutral and Acidic Forms of Cannabinoids Using GC-MS Analysis with Derivatization

#### 2.6.1. Sample Preparation

Fifty µL of the IS mix were added to 100 µL of solutions of CALs and QCs in methanol/cannabis oil previously diluted in methanol 20,000 times. After vortexing, the sample was dried under a gentle nitrogen stream and subsequently reconstituted in 100 µL of BSTFA:TMCS and 30 µL of pyridine. After vortexing, the solution thus reconstituted was incubated at 60 °C for 20 min to obtain derivatization of the analytes through silylation.

#### 2.6.2. GC Conditions

Two μL were injected into the instrument with a split ratio of 5:1. The temperature of the injection port and the transfer line was set at 280 °C. A DB-5MS UI capillary column (30 m × 0.25 mm I.D., 0.25 μm film thickness, with 5% phenylmethylsiloxane) (Agilent Technologies, Milan, Italy) was used to achieve separation. Helium, the carrier gas, was set at a constant flow of 1 mL/min for the total run time (15 min). Oven temperature was initially set at 200 °C and held for 2 min. Then, the temperature rose with a temperature ramp of 20 °C/min until 300 °C; this temperature was held for 6 min. Lastly, the temperature rose with a temperature ramp of 30 °C/min until 325 °C.

#### 2.6.3. MS Conditions

The temperatures of the ion source and the quadrupole were set at 230 °C and 150 °C, respectively. The electron ionization mode was positive. The solvent delay was set at 21 min. The mass spectrometer operated in SIM. As such, the ions monitored were 386 and 371 for Δ9-THC; 390 and 337 for CBD; 487 and 488 for THCA; 453 and 491 for CBDA; 374 for Δ9-THC-d3; and 393 for CBD-d3. The dwell time was set at 70 ms.

### 2.7. Quantification of Neutral and Acidic Forms of Cannabinoids Using LC-MS Analysis

#### 2.7.1. Sample Preparation

Twenty-five µL of the IS mix were added to 50 µL of solutions of CALs and QCs in methanol and 925 µL of 0.1% formic acid and methanol (75:25). The samples were then vortexed.

#### 2.7.2. LC Conditions

Ten µL were injected into the instrument. To achieve chromatographic separation, mobile phases A (0.1% formic acid) and B (methanol) were eluted on a reverse-phase Hypersil GOLD™ C18 (50 mm × 2.1 mm, 1.9 μm; Thermo Scientific™, Monza, Italy) at a constant flow rate of 0.4 mL/min using the following program: 0–1 min, isocratic 25% B; 1–8 min, linear gradient 100% B; 8–11 min, isocratic 100% B; 11–11.1 min, linear gradient 25% B; and 11.1–15 min, isocratic 25% B. The column oven was set at 30 °C. The total run time was 15 min.

#### 2.7.3. MS Conditions

The samples were introduced to the mass spectrometer and ionized positively (Δ9-THC, CBD, and their deuterated analogues) or negatively (THCA and CBDA) via electrospray ionization using the following conditions: curtain gas, 25 psig; collision gas, medium; ion spray voltage, 4500 V for the positive mode and −4500 V for the negative mode; capillary temperature, 450 (°C); and ion source gas, 55 psig.

A multiple reaction monitoring (MRM) mode was adopted. The compound-dependent parameters were previously optimized and used for the analysis. Particularly, the m/z ratio of the precursor ion and product ions (quantifier and qualifier), declustering potential (DP), entrance potential (EP), collision energy (CE), and collision cell exit potential (CXP) for each compound are reported in Table 1.

### 2.8. Data Processing

MassHunter Qualitative and MassHunter Quantitative Analysis software (version 10.2; Agilent Technologies, Milan, Italy) was used for GC-MS data acquisition, visualization, and interpretation. Meanwhile, Analyst (version 1.7) and Multiquant (version 3.0.2) software (Sciex, Milan, Italy) was used for the analysis and processing of LC-MS data.

Since GC-MS with derivatization and LC-MS analysis were used as the “gold standard” methods for comparison, validation of these methods was also performed using the International Council for Harmonisation of Technical Requirements for Registration of Pharmaceuticals for Human Use (ICH) guidelines (EMA/CHMP/ICH/82072/2006, revised on 14 December 2023).

Compound identification was allowed by monitoring the quantifier and qualifier ions at the specific retention time of the analyte. Furthermore, the qualifier ratio, intended as the ratio between the responses of the quantifier and qualifier ions, was checked to be constant. Analyte response derived from the GC-MS and LC-MS analyses was normalized on the IS response. Calibration curves were fit by linear regression with weighting by 1/χ^2^, without forcing the line through the origin.

Concentration of THCA and CBDA in the 3 levels of QCs were calculated by interpolation with the calibration curve relating to the corresponding acidic form in the LC-MS analysis and the GC-MS analysis comprising the derivatization step. Subsequently, the total Δ9-THC and CBD concentration was calculated by multiplying the obtained acidic cannabinoid concentration by 0.877, a factor which accounts for the difference in molecular weight between the acidic and the neutral forms. Regarding the GC-MS analysis not involving the derivatization step, the total Δ9-THC and CBD concentration derived from the decarboxylated forms in the 3 QCs were calculated directly using the analytical method. The yield of the total Δ9-THC and CBD concentrations derived from the corresponding acidic forms was expressed as the percentage of the ratio between the measured and nominal Δ9-THC and CBD concentrations. Measurements were performed using each analytical method on the same batch of QCs in 3 analytical sessions.

The total Δ9-THC and CBD concentrations in real samples of cannabis oil were obtained by interpolation of the analyte response with the calibration curves, conversion of acidic forms to neutral ones (if necessary), and, lastly, by multiplying by the dilution factor of cannabis oils (20,000 times). In particular, calibration curves were capable of measuring cannabinoids from a concentration of 1 to 20 mg/mL.

## 3. Results

### 3.1. Analytical Methods Development

#### 3.1.1. GC-MS Analysis Without Derivatization

The GC-MS method not involving derivatization allowed the detection and quantification only of neutral forms of cannabinoids, derived from the compounds themselves present in the sample and the corresponding acidic forms decarboxylated due to the high temperature of the injection port (280 °C). As such, chromatographic separation was achieved for Δ9-THC and CBD (retention time: 23.4 and 22.5 min, respectively), as well as their IS (retention time: 23.4 and 22.5 min for Δ9-THC-d3 and CBD-d3, respectively) (Figure 1 and Figure 2). Additionally, a linear relationship between the normalized response and the nominal concentration of the analytes was obtained (R^2^ > 0.99) (Figure 1 and Figure 2).

#### 3.1.2. GC-MS Analysis with Derivatization

The GC-MS method involving derivatization allowed the detection and quantification of trimethylsilyl (TMS) derivatives of all forms of cannabinoids, neutral and acidic ones. Chromatographic separation was achieved for Δ9-THC and CBD (retention time: 6.5 and 5.8 min, respectively), THCA and CBDA (retention time: 7.8 and 7.0 min, respectively), as well as their IS (retention time: 6.5 and 5.8 min for Δ9-THC-d3 and CBD-d3, respectively) (Figure 3, Figure 4, Figure 5 and Figure 6). Additionally, a linear relationship between the normalized response and the nominal concentration of the analytes was obtained at least in 3 calibration curves (R^2^ > 0.99) (Figure 3, Figure 4, Figure 5 and Figure 6). Furthermore, the lower limit of detection and quantification of the method correspond to 15 and 50 ng/mL, respectively, for both acidic and neutral cannabinoids. The percentages of intra-day and inter-day accuracy did not exceed 100 ± 13% and 100 ± 14%, respectively. The coefficient of variation demonstrated good intra- and inter-day reproducibility (<12%).

#### 3.1.3. LC-MS Analysis

The LC-MS method allowed the detection and quantification of both neutral and acidic forms of cannabinoids. Chromatographic separation was achieved for Δ9-THC and CBD (retention time: 6.8 and 6.2 min, respectively), THCA and CBDA (retention time: 7.2 and 6.3 min, respectively), as well as their IS (retention time: 6.8 and 6.2 min for Δ9-THC-d3 and CBD-d3, respectively) (Figure 7, Figure 8, Figure 9 and Figure 10). Additionally, a linear relationship between the normalized response and the nominal concentration of the analytes was obtained at least in 3 calibration curves (R^2^ > 0.99) (Figure 7, Figure 8, Figure 9 and Figure 10). Furthermore, the lower limit of detection of the method corresponds to 1 and 3 ng/mL for acidic and neutral cannabinoids, respectively, while the lower limit of quantification corresponds to 5 and 10 ng/mL for acidic and neutral cannabinoids, respectively. The percentages of intra-day and inter-day accuracy did not exceed 100 ± 6% and 100 ± 7%, respectively. The coefficient of variation demonstrated good intra and inter-day reproducibility (<15%).

### 3.2. Total Δ9-THC and CBD Yield in the Laboratory Samples Containing THCA and CBDA

After developing the analytical methods, 3 levels of THCA and CBDA concentration of QCs were tested to obtain the total Δ9-THC and CBD concentration and calculate the yield of recovery of neutral cannabinoids. Notably, when QCs were analyzed using GC-MS without derivatization, the measured concentrations of the total Δ9-THC and CBD were greatly decreased in comparison to the nominal concentration values (Table 2). In particular, the yield of the total Δ9-THC and CBD recovery ranged from 56.1% to 65.5% and from 46.6% to 57.0%, respectively. Instead, the calculated concentration of neutral cannabinoids corresponded to the nominal one if quantification was performed using GC-MS with derivatization or LC-MS (Table 3 and Table 4). In particular, the percentage of yield of the total Δ9-THC and CBD recovery did not exceed 100 ± 17%, a value that falls within the experimental variability according to the ICH guidelines (EMA/CHMP/ICH/82072/2006, revised on 14 December 2023).

### 3.3. Total Δ9-THC and CBD Yield in Real Cannabis Oil Samples

Quantitative results related to the total Δ9-THC and CBD concentrations in cannabis oils obtained using GC-MS analyses with or without the derivatization step are reported in Table 5. As shown, cannabis oils contained both neutral and acidic cannabinoids. Notably, percentages of variation of the quantitative results did not exceed 10% when acidic forms were present in the cannabis oil sample.

## 4. Discussion

*C. sativa* preparations are known to be effective for the psychotropic, analgesic, anti-inflammatory, and antioxidant effects of the main active compounds (cannabinoids); among these galenic preparations, cannabis oil was of interest because of its easy dose adjustment and bioavailability of active molecules in the matrix [20]. As regulated by the Ministerial Decree of 9/11/2015, titration of the active compounds Δ9-THC and CBD has to be performed using sensitive and specific techniques, such as GC-MS and LC-MS, to ensure the quality of the galenic preparation and to ameliorate therapy personalization [4]. Indeed, analytical assessment is fundamental to avoid efficacy and safety implications in patients undergoing oil extract administration [30].

As previously reported in other works, this study presented GC-MS and LC-MS methods for the quantification not only of the active neutral cannabinoids Δ9-THC and CBD, but also of their precursors, the acidic cannabinoids THCA and CBDA, in cannabis oil [3,24]. Comparing the analytical techniques used, LC-MS proved to be the ideal one because of its short sample preparation and the capability to discriminate between neutral and acidic cannabinoids [3,26]. Recently, a cross-validation study by 3 regional reference laboratories in Northern Italy ascertained the interlaboratory reproducibility of cannabis oil quantifications using an LC-MS technique, ensuring high-quality standards and interchangeability between the reference laboratories [3]. Additionally, at our laboratory, the developed LC-MS method showed optimal linearity, sensitivity, reproducibility, and accuracy of the results on samples with known concentrations. However, the main disadvantage of the LC-MS technique is the inability to detect volatile and non-polar compounds in cannabis oil, such as terpenes [26,31].

Furthermore, GC-MS analysis could also be efficient in identifying both neutral and acidic cannabinoids when derivatization occurs [24]. The GC-MS method involving the derivatization step, developed and used in our laboratory, also showed optimal linearity, sensitivity, reproducibility, and accuracy of results on samples with known concentrations, as noticed for the LC-MS method. Even if the sample preparation lengthens, derivatization not only makes cannabinoids more volatile than they originally are, but also ameliorates the peak shape and, importantly, preserves the cannabinoids’ structure [28,32]. Indeed, even though their work was published several years ago and used outdated methods, Dussy et al. for the first time evidenced an incomplete conversion of the acidic form THCA in Δ9-THC caused by the high temperature of the GC injector [28]. In particular, the authors described that the maximum yield of conversion was reached at the temperature of 150 °C; instead, at a higher temperature, Δ9-THC was assumed to be oxidized to cannabinol and other polymeric material [28]. Unfortunately, in the previously mentioned manuscript, only one solution was tested, which was very concentrated, even if concentration seems to play a key role in the decarboxylative process [22,28]. Moreover, no information was provided on CBDA conversion in the study of Dussy et al. We hypothesize that it meets the same fate, as other studies of the decarboxylation kinetics in *C. sativa* hemp suggested [20,22,29].

Given the paucity of scientific literature regarding this topic, we investigated better the thermal conversion of non-derivatized THCA and CBDA in Δ9-THC and CBD during injection in GC-MS systems. Derivatization through silylation is considered an important reaction for hydroxyl protection during synthetic transformations; therefore, acidic cannabinoids not presenting the addition of trimethylsilyl groups are prone to undergo loss of carboxylic groups due to high temperature [27]. Even though this conversion has been largely acknowledged, criticism about the rate of conversion of carboxylates into their products and the total yield of recovery of their derivatives has been raised [27]. Based on our results, and in line with the previous work of Dussy et al., the yield of decarboxylation of non-silylated THCA and CBDA is not complete, resulting in a lower yield of recovery of Δ9-THC and CBD in comparison with the one obtained with LC-MS and GC-MS with derivatization methods. The decreasing rate of conversion was tested in a range of concentrations useful for the quantification of real samples after dilution with similar results (about 50–60% for each cannabinoid) for all the concentrations analyzed. Notably, this suggests that laboratories quantifying the total Δ9-THC and CBD using GC-MS without derivatization would get a lower value than those which use LC-MS or GC-MS with derivatization.

Starting from this finding and based on scientific literature assessing wide variability in the quantification of cannabis oil, we compared the quantitative results obtained using GC-MS methods with or without derivatization [20]. Interestingly, the quantitative results did not exceed the limit fixed for experimental variability when acidic forms were present in the cannabis oil sample. This result leads to several considerations. On the one hand, the main discrepancies in the quantitative results of cannabis oils reported in the literature could be due to the extraction methods used, where decarboxylation also has a fundamental role, or to the storage of the sample before analysis (non-optimal conditions of temperature and light) [20,22]. On the other hand, the bias in the accurate measurement of the total Δ9-THC and CBD was overcome because extraction protocols performed by pharmacists require decarboxylation, leading to only a small amount of acidic forms [20,23]. The results would differ if decarboxylation in the phase of extraction of galenic preparation is not efficient and the content of acidic cannabinoids present in the preparation to titrate is higher than that of the corresponding neutral forms. This would inevitably lead to a marked bias in accurate quantification.

Unfortunately, since all the cannabis oils tested were obtained from local pharmacies using the same extraction protocol (*C. sativa* macerated and extracted in olive oil), investigating the impact of excipients in the formulation on the conversion of acidic cannabinoids was not possible, and this represents a limitation of our study. In conclusion, the non-derivatized acidic forms THCA and CBDA are not completely decarboxylated in the neutral Δ9-THC and CBD in the injection port of GC-MS systems, leading to possible safety implications due to the underestimation of the total cannabinoid content. Indeed, when cannabis oils present a large amount of acidic forms, laboratories measuring only neutral cannabinoids using GC-MS without derivatization report a lower content of the total Δ9-THC and CBD than those which measure cannabinoids with the other methods described. Based on this result, the dose of cannabinoids administered to patients is higher than the one expected when the concentration is underestimated. However, if the galenic preparation is appropriately prepared according to the SIFAP procedure, this scenario is unlikely. Hence, this work aims to advise and raise awareness in pharmacists about the importance of the decarboxylative process during cannabis oil preparation.

## Figures and Tables

**Figure 1 pharmaceutics-17-00334-f001:**
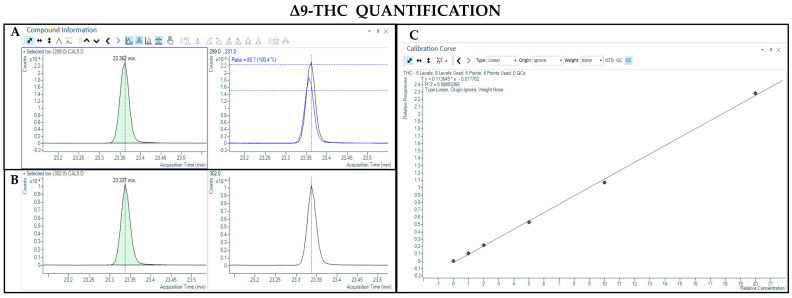
Chromatograms obtained by GC-MS analysis without derivatization and the qualifier ratio related to Δ9-THC (**A**) and its deuterated analogue (Δ9-THC-d3) used as the IS (**B**). Calibration curve built for Δ9-THC quantification (**C**).

**Figure 2 pharmaceutics-17-00334-f002:**
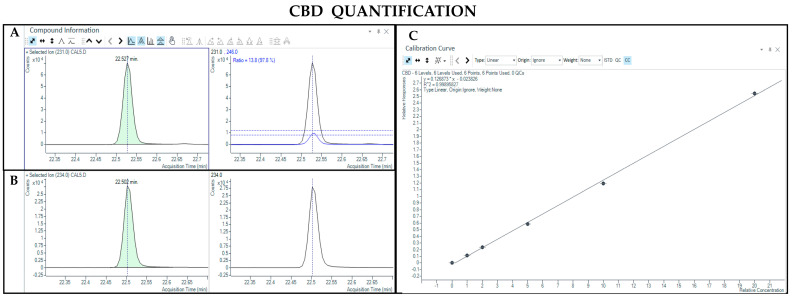
Chromatograms obtained by GC-MS analysis without derivatization and the qualifier ratio related to CBD (**A**) and its deuterated analogue (CBD-d3) used as the IS (**B**). Calibration curve built for CBD quantification (**C**).

**Figure 3 pharmaceutics-17-00334-f003:**
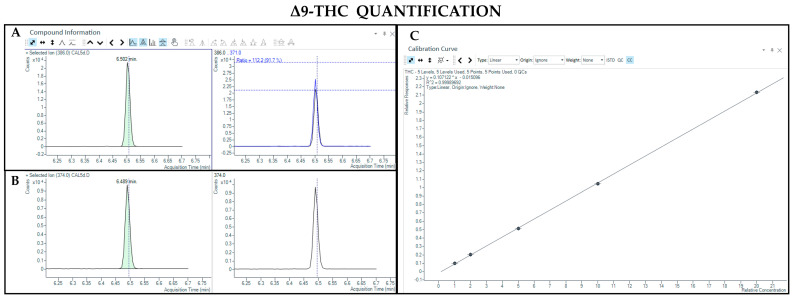
Chromatograms obtained by GC-MS analysis with derivatization and the qualifier ratio related to Δ9-THC (**A**) and its deuterated analogue (Δ9-THC-d3) used as the IS (**B**). Calibration curve built for Δ9-THC quantification (**C**).

**Figure 4 pharmaceutics-17-00334-f004:**
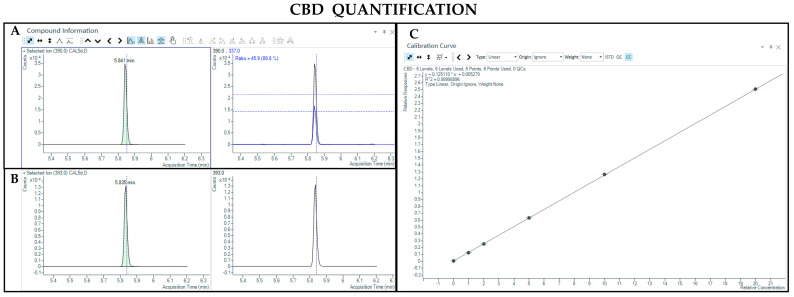
Chromatograms obtained by GC-MS analysis with derivatization and the qualifier ratio related to CBD (**A**) and its deuterated analogue (CBD-d3) used as the IS (**B**). Calibration curve built for CBD quantification (**C**).

**Figure 5 pharmaceutics-17-00334-f005:**
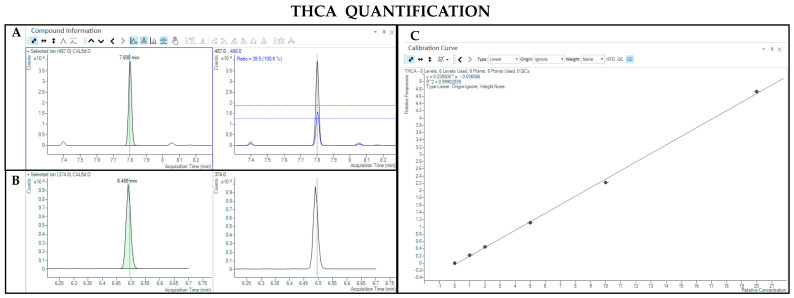
Chromatograms obtained by GC-MS analysis with derivatization and the qualifier ratio related to THCA (**A**) and Δ9-THC-d3 used as the IS (**B**). Calibration curve built for THCA quantification (**C**).

**Figure 6 pharmaceutics-17-00334-f006:**
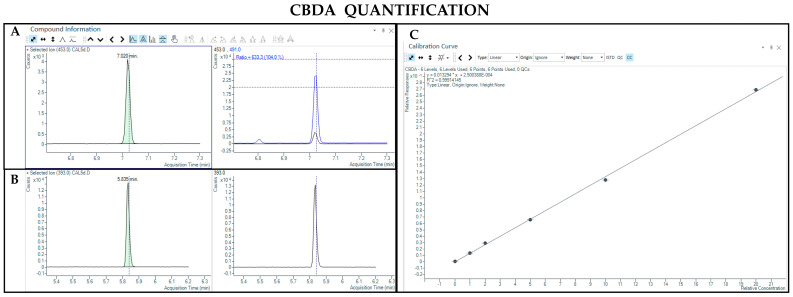
Chromatograms obtained by GC-MS analysis with derivatization and the qualifier ratio related to CBDA (**A**) and CBD-d3 used as the IS (**B**). Calibration curve built for CBDA quantification (**C**).

**Figure 7 pharmaceutics-17-00334-f007:**
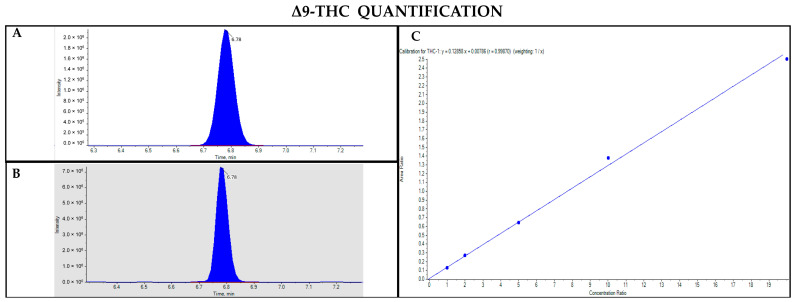
Chromatograms obtained by LC-MS analysis related to Δ9-THC (**A**) and its deuterated analogue (Δ9-THC-d3) used as the IS (**B**). Calibration curve built for Δ9-THC quantification (**C**).

**Figure 8 pharmaceutics-17-00334-f008:**
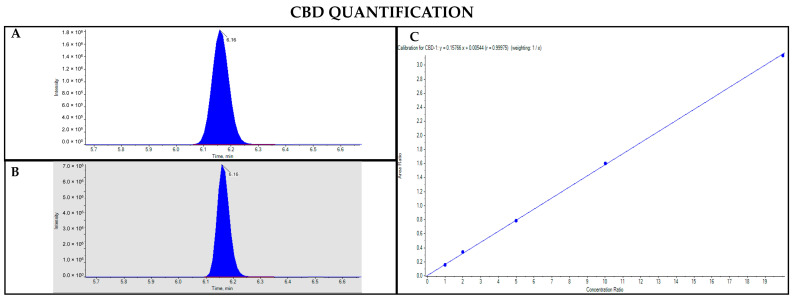
Chromatograms obtained by LC-MS analysis related to CBD (**A**) and its deuterated analogue (CBD-d3) used as the IS (**B**). Calibration curve built for CBD quantification (**C**).

**Figure 9 pharmaceutics-17-00334-f009:**
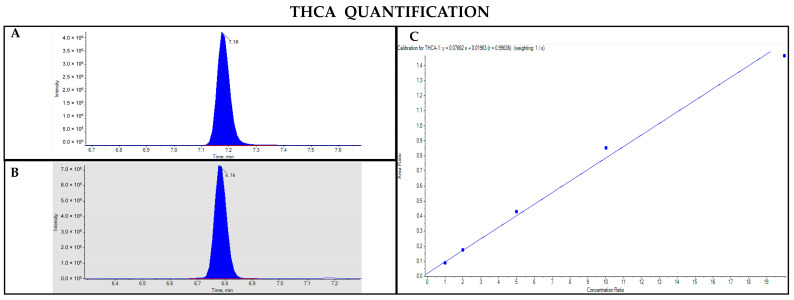
Chromatograms obtained by LC-MS analysis related to THCA (**A**) and Δ9-THC-d3 used as the IS (**B**). Calibration curve built for THCA quantification (**C**).

**Figure 10 pharmaceutics-17-00334-f010:**
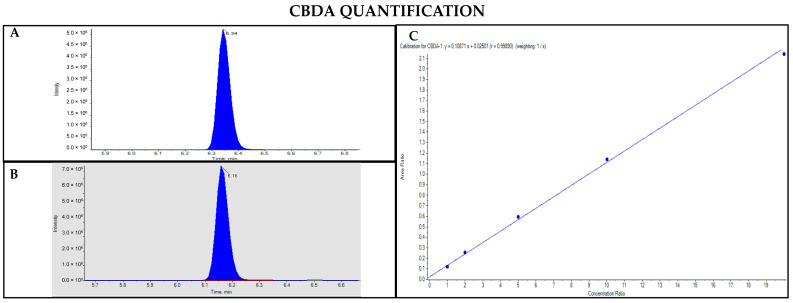
Chromatograms obtained by LC-MS analysis related to CBDA (**A**) and CBD-d3 used as the IS (**B**). Calibration curve built for CBDA quantification (**C**).

**Table 1 pharmaceutics-17-00334-t001:** Compound-dependent parameters. DP, declustering potential; EP, entrance potential; CE, collision energy; CXP, collision cell exit potential.

Compound	Precursor Ion (*m*/*z*)	Product Ion (*m*/*z*)	DP	EP	CE	CXP
THC	315.3	193.1	50	6	30	12
		123.1	50	6	40	11
CBD	315.3	193.1	50	6	30	12
		123.1	50	6	40	11
THC-d3	318.3	196.1	50	6	42	15
CBD-d3	318.3	196.1	50	6	42	15
THCA	357.1	339.0	−115	−10	−29	−20
		245.0	−115	−10	−40	−20
CBDA	357.1	339.0	−115	−10	−29	−20
		245.0	−115	−10	−40	−20

**Table 2 pharmaceutics-17-00334-t002:** Total Δ9-THC and CBD concentrations measured using GC-MS analysis without derivatization and the yield of the quantification process. SD, standard deviation.

GC-MS WITHOUT DERIVATIZATION						
Standard	Total Δ9-THC, Nominal Concentration (ng/mL)	Total Δ9-THC, Calculated Concentration (ng/mL) ± SD	Total CBD, Nominal Concentration (ng/mL)	Total CBD, Calculated Concentration (ng/mL) ± SD	Yield, Total Δ9-THC (%)	Yield, Total CBD (%)
QCI	175	108 ± 0.01	175	92 ± 0.003	61.8	52.3
QCII	351	228 ± 0.03	351	200 ± 0.01	65.5	57.0
QCIII	702	393 ± 0.04	702	327 ± 0.03	56.1	46.6

**Table 3 pharmaceutics-17-00334-t003:** Total Δ9-THC and CBD concentrations measured using GC-MS analysis with derivatization and the yield of the quantification process. SD, standard deviation.

GC-MS WITH DERIVATIZATION						
Standard	Total Δ9-THC, Nominal Concentration (ng/mL)	Total Δ9-THC, Calculated Concentration (ng/mL) ± SD	Total CBD, Nominal Concentration (ng/mL)	Total CBD, Calculated Concentration (ng/mL) ± SD	Yield, Total Δ9-THC (%)	Yield, Total CBD (%)
QCI	175	175 ± 0.03	175	196 ± 0.04	100.0	111.7
QCII	351	349 ± 0.02	351	393 ± 0.06	99.6	112.1
QCIII	702	722 ± 0.09	702	757 ± 0.06	102.9	107.9

**Table 4 pharmaceutics-17-00334-t004:** Total Δ9-THC and CBD concentrations measured using LC-MS analysis and the yield of the quantification process. SD, standard deviation.

LC-MS						
Standard	Total Δ9-THC, Nominal Concentration (ng/mL)	Total Δ9-THC, Calculated Concentration (ng/mL) ± SD	Total CBD, Nominal Concentration (ng/mL)	Total CBD, Calculated Concentration (ng/mL) ± SD	Yield, Total Δ9-THC (%)	Yield, Total CBD (%)
QCI	175	191 ± 0.01	175	196 ± 0.09	109.0	111.9
QCII	351	404 ± 0.01	351	409 ± 0.02	115.2	116.5
QCIII	702	715 ± 0.01	702	714 ± 0.03	101.9	101.8

**Table 5 pharmaceutics-17-00334-t005:** Total Δ9-THC and CBD concentrations in cannabis oil samples measured using GC-MS analyses with or without derivatization. (a) Presence of a small amount of acidic form of the corresponding cannabinoid.

Inflorescence Variety	GC-MS Analysis with Derivatization		GC-MS Analysis Without Derivatization		Variation (%)	
	Δ9-THC (mg/mL)	CBD (mg/mL)	Δ9-THC (mg/mL)	CBD (mg/mL)	Δ9-THC	CBD
Bedrocan	13.4 (a)	ND	12.9	ND	4%	ND
Bediol, No. 1	4.7	7.1 (a)	4.9	6.5	4%	8.5%
Bediol, No. 2	4.8	6.9 (a)	4.7	6.5	2%	5.8%
Bediol, No. 3	5.1	7.5 (a)	5.2	7.1	2%	5.4%
Bediol, No. 4	4.8	7.3 (a)	4.8	6.6	0%	10%
Bediol, No. 5	5.1	7.5 (a)	4.9	6.9	4%	8%

## Data Availability

Data are available from the corresponding author upon reasonable request.

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
