# Peer review of "Incomplete Decarboxylation of Acidic Cannabinoids in GC-MS Leads to Underestimation of the Total Cannabinoid Content in Cannabis Oils Without Derivatization"

_pharmaceutics, 2025, doi:10.3390/pharmaceutics17030334_

Round 1

Reviewer 1 Report

Comments and Suggestions for Authors

The United Nations Office on Drugs and Crime (UNODC) recommends using Gas Chromatography-Mass Spectrometry (GC-MS) and Liquid Chromatography-Mass Spectrometry (LC-MS) for the identification and analysis of cannabis and its products. According to the guidelines, complete decarboxylation is advised prior to GC-MS analysis to accurately assess neutral cannabinoids. In contrast, LC-MS, accompanied by derivatization, is suggested for the comprehensive analysis of both neutral and acidic cannabinoids.

This research method shows limited novelty, and the identified research gap lacks substantial interest. The incomplete conversion from acidic to neutral cannabinoids may result from the presence of other excipients in cannabis oil or within commercial cannabis-containing products. Expanding the study to include a broader range of samples to investigate the impact of excipients in cannabis formulations on the conversion of acidic cannabinoids could enhance its interest and relevance.

The analytical method should provide comprehensive information on system suitability and method validation results.

Please illustrate the derivatization reaction and provide an explanation of how the derivatizing agent can safeguard against the acidic conversion of cannabinoids.

Please provide more detailed information in the conclusion section, particularly regarding the results derived from your findings.

Minor comments:

  1. Figure 1-10 shows that the calibration curves have low visibility.
  2. Line 289, Table 3-4: Is the recovery of ±17% acceptable for the analysis of Δ9-THC and CBD standards?
  3. Line 303 should refer to Table 5, not Table 3.

Author Response

Comment: The United Nations Office on Drugs and Crime (UNODC) recommends using Gas Chromatography-Mass Spectrometry (GC-MS) and Liquid Chromatography-Mass Spectrometry (LC-MS) for the identification and analysis of cannabis and its products. According to the guidelines, complete decarboxylation is advised prior to GC-MS analysis to accurately assess neutral cannabinoids. In contrast, LC-MS, accompanied by derivatization, is suggested for the comprehensive analysis of both neutral and acidic cannabinoids.

This research method shows limited novelty, and the identified research gap lacks substantial interest. The incomplete conversion from acidic to neutral cannabinoids may result from the presence of other excipients in cannabis oil or within commercial cannabis-containing products. Expanding the study to include a broader range of samples to investigate the impact of excipients in cannabis formulations on the conversion of acidic cannabinoids could enhance its interest and relevance.

Answer: We thank the reviewer for the comment. Even if complete decarboxylation is advised according to guidelines, cannabis oil samples realistically present acidic cannabinoids according to our laboratory experience justifying our study. Based on the results obtained, our work also supported the use of LC-MS as the gold standard method for cannabinoids quantification. Furthermore, we agree with the reviewer that a limitation of the study consists in not investigating the impact of excipients on the conversion of acidic cannabinoids into their neutral forms and we specified this limitation at the end of the discussion.

Comment: The analytical method should provide comprehensive information on system suitability and method validation results.

Answer: We thank the reviewer for the comment and we modified the manuscript including briefly the main validation parameters of the analytical methods used for comparison with the GC-MS method not involving derivatization in the results section (paragraph 3.1).

Comment: Please illustrate the derivatization reaction and provide an explanation of how the derivatizing agent can safeguard against the acidic conversion of cannabinoids.

Answer: We thank the reviewer for the suggestion and we provide a clearer explanation of how the silylation can safeguard the decarboxylation of acidic cannabinoids (introduction: lines 91-95; discussion: lines 391-399).

Comment: Please provide more detailed information in the conclusion section, particularly regarding the results derived from your findings.

Answer: We thank the reviewer for the suggestion and we ameliorated the conclusion of our work including the main finding and limitations.

Minor comments:

  1. Figure 1-10 shows that the calibration curves have low visibility.

Answer: We thank the reviewer for the comment. As the image resolution inevitably become worse in the manuscript word file, we uploaded also a pdf comprising all the figures present in the manuscript.

  1. Line 289, Table 3-4: Is the recovery of ±17% acceptable for the analysis of Δ9-THC and CBD standards?

Answer: We thank the reviewer for the comment. We specified that the recovery of 100±17% of analyte could be considered a value that falls within the experimental variability as ICH guidelines considered percentages of accuracy of 80% and 120% of the declared content as suggested lower and higher threshold.

  1. Line 303 should refer to Table 5, not Table 3.

Answer: We thank the reviewer for the comment and we modified the manuscript accordingly.

Reviewer 2 Report

Comments and Suggestions for Authors

Manuscript ID: pharmaceutics-3477918

Title: Incomplete decarboxylation of non-derivatized acidic forms of cannabinoids in GC-MS: advice in the use of cannabis oil

The study investigates conversion of acidic into neutral forms of Δ9-THC and CBD during analytical determination, and its influence on quantification of active cannabinoids in galenic preparations. The decarboxylative process during cannabis oil preparation is of significant importance for pharmacists, and awareness of possible influence of the analytical methods applied for the quality control is important for mitigation of the risk of underestimation of total cannabinoid content.

The topic of the article is relevant for the field of pharmaceutical science and fits the journal scope. The article is well-structured and neatly written. The abstract clearly and accurately reflects presented data. The introductory section provides adequate scientific background for the study. The authors address well-defined questions (whether the conversion of acidic into neutral forms, occurring during analytical determination, can influence quantification of active cannabinoids Δ9-THC and CBD and thus affect galenic preparation quality; what is the conversion rate), utilizing sound and technically solid experimental design (comparison of GC-MS with or without derivatization and LC-MS analyses at known concentration of acidic forms in reference samples and real cannabis oil samples). The methods of analysis (GC-MS with derivatization, GC-MS without derivatization, LC-MS method) are relevant. i.e., used in testing laboratories, appropriately validated according to internationally recognized protocols, and described with sufficient details. The study results are aligned with the main research aims and described accurately. The results are robust enough and interpreted correctly and consistently throughout the article, including tabular and graphical representations, which are clear and useful. The obtained results are appropriately commented considering the context provided by previously published studies of relevance. Research is supported by sufficient data, providing valid and useful conclusions, consistent with the evidence and arguments presented in the article. Cited references are appropriate and relevant, mostly recent publications, with only one self-citation.  

Specific comments:

Figures: considering that figures (as well as tables) should be self-explanatory, it is advisable to mark which type of chromatogram is presented - identify the method in the figure caption, e.g., GC-MS (relevant for Figures 1-10).

Tabel 5: in the Variation column, units of measurement, mg/ml (second heading row) should be deleted for both Δ9-THC and CBD

Comments on the Quality of English Language

The English could be improved to more clearly express the research, i.e., to avoid ambiguity in some sentences and possible confusion among readers who are not familiar with applied analytical techniques

Author Response

Comment: The study investigates conversion of acidic into neutral forms of Δ9-THC and CBD during analytical determination, and its influence on quantification of active cannabinoids in galenic preparations. The decarboxylative process during cannabis oil preparation is of significant importance for pharmacists, and awareness of possible influence of the analytical methods applied for the quality control is important for mitigation of the risk of underestimation of total cannabinoid content.

The topic of the article is relevant for the field of pharmaceutical science and fits the journal scope. The article is well-structured and neatly written. The abstract clearly and accurately reflects presented data. The introductory section provides adequate scientific background for the study. The authors address well-defined questions (whether the conversion of acidic into neutral forms, occurring during analytical determination, can influence quantification of active cannabinoids Δ9-THC and CBD and thus affect galenic preparation quality; what is the conversion rate), utilizing sound and technically solid experimental design (comparison of GC-MS with or without derivatization and LC-MS analyses at known concentration of acidic forms in reference samples and real cannabis oil samples). The methods of analysis (GC-MS with derivatization, GC-MS without derivatization, LC-MS method) are relevant. i.e., used in testing laboratories, appropriately validated according to internationally recognized protocols, and described with sufficient details. The study results are aligned with the main research aims and described accurately. The results are robust enough and interpreted correctly and consistently throughout the article, including tabular and graphical representations, which are clear and useful. The obtained results are appropriately commented considering the context provided by previously published studies of relevance. Research is supported by sufficient data, providing valid and useful conclusions, consistent with the evidence and arguments presented in the article. Cited references are appropriate and relevant, mostly recent publications, with only one self-citation.  

Answer: We thank the reviewer for the positive comment.

Specific comments:

Figures: considering that figures (as well as tables) should be self-explanatory, it is advisable to mark which type of chromatogram is presented - identify the method in the figure caption, e.g., GC-MS (relevant for Figures 1-10).

Answer: We agree with the reviewer and we modified the manuscript accordingly.

Tabel 5: in the Variation column, units of measurement, mg/ml (second heading row) should be deleted for both Δ9-THC and CBD

Answer: We agree with the reviewer and we modified the manuscript accordingly.

Comments on the Quality of English Language

The English could be improved to more clearly express the research, i.e., to avoid ambiguity in some sentences and possible confusion among readers who are not familiar with applied analytical techniques

Answer: We thank the reviewer for the comment and we went through all the manuscript to improve the quality of English language.

Reviewer 3 Report

Comments and Suggestions for Authors

The manuscript entitled: “Incomplete decarboxylation of non-derivatized acidic forms of cannabinoids in GC-MS: advice in the use of cannabis oil” deals with determination and quantification of neutral and acidic cannabinoids using GC-MS and comparing with LC-MS. The results are presented and discussed in regard to prior derivatization of samples and in the absence of the derivatization step.

The main issue with the manuscript is a lack of clearly stated premise of the work. The Title itself provides no explicit information regarding of the most relevant obtained results in regard to the experimental procedure. Therefore, is should be changed to better to better present the key finding of the research.

In the manuscript, new GC-Method is mentioned for the analysis, but not much is stated about the method itself, the novelty or advantages. The authors have published a paper in reference 24 on this subject but it is not even cited in GC-MS methodology.

Furthermore, the importance of connection between decarboxylation and derivatization should be better elaborated. For example:

In the abstract it is stated: “Non-derivatized THCA/CBDA are not completely decarboxylated in Δ9-THC/CBD in GC-MS systems due to high temperature …” but later in the Introduction it is stated “… with a derivatization step, which is needed not only to make cannabinoids more volatile but also to avoid the decarboxylation of acidic forms due to the high temperature of the injection port”

And then in the discussion: In line with the above-discussed works, the yield of decarboxylation of THCA and CBDA, intended also as the yield of recovery of Δ9-THC and CBD, was greatly decreased when samples were not derivatized.

The sentence in line 53 should be rewritten, it is unclear.

Are Bedrocan, Bediol…etc. Canabis varieties or brand names for cultivar/medicinal preparations?

Line 284, beginning of the sentence… English language should be corrected.

In paragraph 3.3. Table 5 instead of Table 3.

Comments on the Quality of English Language

On occasions sentences are unclear and Grammar should be checked.

Author Response

Comment: The manuscript entitled: “Incomplete decarboxylation of non-derivatized acidic forms of cannabinoids in GC-MS: advice in the use of cannabis oil” deals with determination and quantification of neutral and acidic cannabinoids using GC-MS and comparing with LC-MS. The results are presented and discussed in regard to prior derivatization of samples and in the absence of the derivatization step.

The main issue with the manuscript is a lack of clearly stated premise of the work. The Title itself provides no explicit information regarding of the most relevant obtained results in regard to the experimental procedure. Therefore, is should be changed to better to better present the key finding of the research.

Answer: We thank the reviewer for the comment and we ameliorated the manuscript changing the title to better state the premise of our work.

In the manuscript, new GC-Method is mentioned for the analysis, but not much is stated about the method itself, the novelty or advantages. The authors have published a paper in reference 24 on this subject but it is not even cited in GC-MS methodology.

Answer: We thank the reviewer for the comment. Reference 24 reported a previous method used in our laboratory. To date, new methods (GC-MS with derivatization and LC-MS methods) were developed and validated. We mentioned briefly the validation parameters in the results section (paragraph 3.1) and evidenced the advantages of the new method in the discussion section (lines 366-377).

Furthermore, the importance of connection between decarboxylation and derivatization should be better elaborated. For example:

In the abstract it is stated: “Non-derivatized THCA/CBDA are not completely decarboxylated in Δ9-THC/CBD in GC-MS systems due to high temperature …” but later in the Introduction it is stated “… with a derivatization step, which is needed not only to make cannabinoids more volatile but also to avoid the decarboxylation of acidic forms due to the high temperature of the injection port”

And then in the discussion: In line with the above-discussed works, the yield of decarboxylation of THCA and CBDA, intended also as the yield of recovery of Δ9-THC and CBD, was greatly decreased when samples were not derivatized.

Answer: We thank the reviewer for the suggestion and we provide a clearer explanation of the connection between decarboxylation and derivatization (introduction: lines 91-95; discussion: lines 391-399).

The sentence in line 53 should be rewritten, it is unclear.

Answer: We thank the reviewer for the comment and we rewrote the sentence (lines 62-65).

Are Bedrocan, Bediol…etc. Canabis varieties or brand names for cultivar/medicinal preparations?

Answer: We thank the reviewer for the comment and we specified that Bedrocan, Bediol, etc. are cannabis varieties (lines 62 and 69).

Line 284, beginning of the sentence… English language should be corrected.

Answer: We thank the reviewer for the comment and we corrected the sentence (line 321).

In paragraph 3.3. Table 5 instead of Table 3.

Answer: We thank the reviewer for the comment and we modified the manuscript accordingly.

Comments on the Quality of English Language

On occasions sentences are unclear and Grammar should be checked.

Answer: We thank the reviewer for the comment and we went through all the manuscript to improve the quality of English language.

Round 2

Reviewer 1 Report

Comments and Suggestions for Authors

The manuscript has been revised according to the recommendations.

Reviewer 3 Report

Comments and Suggestions for Authors

The authors have addressed all the issues from the first revision and the manuscript has been significantly improved and can therefore be accepted for publication.